# Immunological Prognostic Factors in Multiple Myeloma

**DOI:** 10.3390/ijms22073587

**Published:** 2021-03-30

**Authors:** Dominika Bębnowska, Rafał Hrynkiewicz, Ewelina Grywalska, Marcin Pasiarski, Barbara Sosnowska-Pasiarska, Iwona Smarz-Widelska, Stanisław Góźdź, Jacek Roliński, Paulina Niedźwiedzka-Rystwej

**Affiliations:** 1Institute of Biology, University of Szczecin, 71-412 Szczecin, Poland; dominika.bebnowska@usz.edu.pl (D.B.); rafal.hrynkiewicz@usz.edu.pl (R.H.); 2Department of Clinical Immunology and Immunotherapy, Medical University of Lublin, 20-093 Lublin, Poland; ewelina.grywalska@gmail.com (E.G.); jacek.rolinski@gmail.com (J.R.); 3Department of Immunology, Faculty of Health Sciences, Jan Kochanowski University, 25-317 Kielce, Poland; marcinpasiarski@gmail.com; 4Department of Hematology, Holy Cross Cancer Centre, 25-734 Kielce, Poland; 5Department of Oncocardiology, Holy Cross Cancer Centre, 25-734 Kielce, Poland; spbasia@gmail.com; 6Department of Nephrology, Cardinal Stefan Wyszynski Provincial Hospital in Lublin, 20-718 Lublin, Poland; i.widelska@interia.pl; 7Faculty of Medicine and Health Sciences, The Jan Kochanowski University, 25-516 Kielce, Poland; stanislawgozdz1@gmail.com; 8Department of Clinical Oncology, Holy Cross Cancer Centre, 25-734 Kielce, Poland

**Keywords:** cytokines, immune cells, markers, multiple myeloma, prognostic factors

## Abstract

Multiple myeloma (MM) is a plasma cell neoplasm characterized by an abnormal proliferation of clonal, terminally differentiated B lymphocytes. Current approaches for the treatment of MM focus on developing new diagnostic techniques; however, the search for prognostic markers is also crucial. This enables the classification of patients into risk groups and, thus, the selection of the most optimal treatment method. Particular attention should be paid to the possible use of immune factors, as the immune system plays a key role in the formation and course of MM. In this review, we focus on characterizing the components of the immune system that are of prognostic value in MM patients, in order to facilitate the development of new diagnostic and therapeutic directions.

## 1. Introduction

Multiple myeloma (MM) is a neoplasm of the hematopoietic system, classified by the World Health Organization (WHO) as a plasma cell neoplasm [1]. MM is characterized by an abnormal proliferation of clonal, terminally differentiated B lymphocytes. Malignant plasma cells are present primarily in the bone marrow, but can also be recorded in peripheral blood and other extramedullary sites, especially in the advanced stage of the disease [2]. In the course of MM, the overproduction of monoclonal proteins (M-proteins) is observed, which then accumulate in the serum and urine. This condition causes the occurrence of extensive disease symptoms, including anaemia, hypercalcemia, and kidney and bone damage, as well as immunosuppression [3]. Despite extensive research, the aetiology of this disease remains unknown. Nevertheless, analyses of epidemiological data have suggested that genetic factors, chronic antigenic stimulation, and environmental and occupational factors play a role in the pathogenesis of MM [4]. It has been estimated that the incidence of MM globally is 2 per 100,000 people, while in Europe it is 6 per 100,000 people [5]. These data make this neoplasm the second-most common haematological malignancy. Importantly, this disease mainly affects people over 65 years of age, where the median age of patients at diagnosis is 70 years old [6]. Despite advances in the development of therapeutic methods, the disease is still incurable in most cases. Mortality due to MM is quite high, and it has been estimated that, in 2020, there were 12,830 deaths in the USA; meanwhile, in Europe, 31,000 patients have been reported to die each year [3]. The current directions of MM research are largely focused on developing diagnostic methods and therapies aimed at improving patient survival. An equally important approach is the search for prognostic markers enabling the classification of patients into risk groups, which has a significant impact on the selection of an appropriate therapeutic approach, as well as on the effectiveness and tolerance of the treatment [7]. The prognostic factors of the course of the disease can take various forms and can be classified into very diverse groups. It cannot be denied that the most expressive factor indicating a worse prognosis is age, as patients over 50 years of age have a significantly shorter survival median than young patients [4]. The immune system plays an essential role in the development and course of MM, where the immune cells present in the tumour environment and the released cytokines significantly influence the disease progression [8]. There are a number of elements in the immune system, the analysis of which allows for obtaining valuable data related to the prognosis of the course of the disease. In this review, we describe immunological markers that have prognostic value in patients with MM, in order to organize the current state of knowledge to aid in the development of new diagnostic and therapeutic directions.

## 2. Cytokines and Chemokines

Cytokines play a fundamental role in modulating the immune response and are often produced by immune cells. MM cells and bone marrow stromal cells (BMSC) present in the tumour environment induce the autocrine or paracrine production of many mediators and largely influence the maintenance of the cytokine feedback loop [9]. In MM, the cytokine network is involved in tumour growth, progression, and spread. It has also been shown that cytokines are involved in the destruction of the bone marrow, which is typically observed in the course of MM [10,11]. It has been reported that the documented activity of multiple cytokine subsets in MM patients, such as interleukins, tumour necrosis factors, growth factors, interferons, and chemokines, can serve as prognostic factors for the course of the disease.

### 2.1. Pro-Inflammatory Cytokines

#### 2.1.1. IL-1

IL-1 is the major cytokine responsible for local and systemic inflammation. It is produced by fibroblasts, monocytes, tissue macrophages, and dendritic cells (DC), as well as B lymphocytes, epithelial cells, and NK cells [12]. It has been shown that IL-1ß in the MM microenvironment stimulates the production of IL-6, which is the main factor in tumour survival and proliferation [13]. Its action, depending on the circumstances, may be beneficial (by supporting innate immunity), but its negative effects have also been observed. The participation of IL-1 is observed under conditions of stress and chronic inflammation, conditions that strongly favour neoplastic development and disease progression. It has also been reported that tumour cells can directly produce IL-1 in a positive feedback loop, contributing to the failure of targeted therapies. Overall, a high level of IL-1 is a factor that leads to poor prognosis in cancer patients [11]. Clinical studies (NCT00635154) have shown that IL-1 plays an important role in the conversion of the latent form of myeloma to the active form of MM [14]. Patients with smouldering multiple myeloma (SMM) or indolent multiple myeloma (IMM) after targeted treatment on the IL-1/IL-6 axis using Anakinra (an IL-1 receptor antagonist; IL-1Ra) showed improvement in PFS (progression-free survival) and overall survival (OS) parameters [14]. Interestingly, the use of the specific IL-1β affinity P2D7KK antibody resulted in a survival rate of 70% in a murine myeloma model [15]. Moreover, studies have shown [9] that the amount of IL-1ß in the serum of MM patients is significantly increased. These data lead to the classification of IL-1 as a valuable prognostic factor for the course of the disease in MM patients.

#### 2.1.2. IL-2

IL-2 is a pro-inflammatory cytokine produced mainly by CD8 + and CD4 + T-cells, which plays a pivotal role in the T-cell response. Target cells that are affected by IL-2, in addition to CD8 + and CD4 + T-cells, are B-cells and CD4 + T-cells. IL-2 strongly stimulates the growth of NK and T cells and enhances their cytolytic effect [12]. The IL-2–IL-2 receptor (IL-2R) system plays a key role in maintaining the proper functioning of T-cells. Abnormal levels of soluble IL-2R (sIL-2R) may affect the imbalance of the IL-2/IL-2R system, resulting in a disturbance of T-cell immunoregulation. It is worth mentioning that the therapeutic use of IL-2 together with TNF-α has been shown to influence the elimination of the tumour in a mouse model of myeloma, emphasizing the importance of IL-2 in anti-cancer therapy [16]. In vitro studies have shown that the γδ lymphocytes generated after treatment with IL-2 and bisphosphates are effective in killing myeloma plasma cells [17]. A phase II study of IL-2 and zoledronate therapy in patients with MM after autologous bone marrow transplant has been described. An increase in the number of γδ lymphocytes, which exhibit anti-cancer properties, was observed after the use of these therapeutic agents; however, ultimately, this therapy had little effect [18]. It is well-known that IL-2–IL-2R has a significant prognostic value in monoclonal gammopathies. Studies have been conducted [19], which have shown an increased value of sIL-2R in the urine and serum of MM patients, compared to the control group. The observed increase in sIL-2R was also related to the activity of the disease. Moreover, the active form of MM showed a significantly higher percentage of IL-2R + plasma cells, compared to the stable form of MM or Monoclonal gammopathy of undetermined significance (MGUS). Thus, a disorder of the IL-2–IL-2R system may be a useful prognostic marker in the diagnosis of MM, in comparison to MGUS, and may be used as a malignancy marker [20]. Other studies have confirmed [21] that the level of sIL-2R in the blood serum is an important marker for predicting the treatment response and is a prognostic factor for PFS. In these studies, high sIL-2R levels correlated with poorer PFS in patients receiving bortezomib-based therapy. Moreover, it has also been shown that the expression of IL-2 membrane receptors on bone marrow plasma cells and peripheral blood mononuclear cells (PBMCs) is correlated with MM activity and disease severity [22].

#### 2.1.3. IL-6

IL-6 is a pleiotropic cytokine that, depending on the signalling pathway, may have pro- and anti-inflammatory effects. It can be produced in response to the stimulation of IL-1 and tumour necrosis factor (TNF), by monocytes, endothelial cells, macrophages, or fibroblasts. The action of IL-6 stimulates the proliferation and activation of T-lymphocytes, differentiation of B-lymphocytes, but also controls the acute phase responses [12]. Increased levels of IL-6 have been observed in many MM patients [23,24], indicating that this cytokine plays an essential role in the development of MM. This has been confirmed by studies in which IL-6 stimulated the in vitro growth of fresh myeloma cells isolated from patients, while another study showed that anti-IL-6 antibodies had anti-tumour effects in MM [25,26]. Siltuximab is an anti-IL-6 monoclonal antibody (mAb). Randomized clinical trials of 286 patients were conducted to analyse the effect of additional silituximab in bortezomib therapy in patients with relapsed refractory MM. As a result, it was not confirmed that the addition of siltuximab improved the PFS and OS of patients, but it was found that this combination treatment led to a shorter response time and numerically longer duration of response than bortezomib alone [27]. As IL-6 involvement is observed in the early stages of the disease, where the myeloma interacts with the stroma and plays a key role in cancer cell survival, anti-IL-6 therapy in the early stages of the disease may have more promising results [25]. On the other hand, a clinical study (NCT01484275) that included 85 patients showed that siltuximab may delay the disease progression in patients with high-risk SMM. However, the study did not obtain satisfactory results, regarding the annual PFS [28]. Thus, a clinical trial (NCT01309412) was initiated to find out whether this agent can reduce the severity of symptoms after autologous stem cell transplant in patients with MM and systemic AL amyloidosis. In contrast, clinical trials (NCT02057770) for the treatment of myeloma using tocilizumab, which is a humanized anti-IL-6R mAb, have been discontinued [29]. Clinical trials of targeted therapies for IL-6 certainly require many more studies, but the role of IL-6 as a prognostic marker for MM patients can be considered undeniable. Moreover, some data have suggested that IL-6 is a factor that determines survival in MM, as it inhibits apoptosis in myeloma cells [26]. Some myeloma cell lines have been shown to be capable of autocrine and paracrine IL-6 secretion, while others do not express IL-6. Nevertheless, it has been suggested that the generation of human MM may be mediated by autocrine IL-6 release by myeloma plasma cells [22], while the production of IL-6 by bone marrow stromal cells may be paracrine [20]. The conducted studies have shown that the levels of IL-6 and sIL-6R in MM patients, both in marrow plasma cells and in peripheral blood, are of great clinical importance [20]. Certain correlations related to these markers have been recorded directly in MM patients [22]. It has been shown that high serum levels of sIL-6R are associated with the early or late phase of the disease, rather than the plateau phase. Moreover, increased levels of sIL-6R directly correlate with shorter survival, compared to the low levels of sIL-6R that have been observed in patients with longer survival [30]. Other studies have indicated that there is a relationship between the sIL-6R level and the tumour weight [31]. This is consistent with the results obtained in the remaining analyses, which showed that elevated levels of IL-6 and sIL-6R reflect the level of disease activity and indicate a poor prognosis. Significantly elevated values of IL-6 in MM patients were observed in patients with advanced disease. At the same time, the decrease in the above-mentioned parameters corresponded to a positive response to the treatment [32]. Many clinical studies have been carried out [33], the results of which confirm that high levels of IL-6 have been found in MM patients which, in turn, enables the differentiation of MM from MGUS, thanks to which an additional diagnostic value has been assigned to this marker. It has been reported that high cell expression of IL-6 mRNA in MGUS patients may predict the development of multiple myeloma. It was also found that, in patients with malignant gammopathies, the level of sIL-6R measured in the serum was significantly higher than in the case of MGUS [22].

#### 2.1.4. IL-8

IL-8, also known as CXCL8, is a pro-inflammatory CXC chemokine [34]. It is involved in many physiological and pathophysiological processes [35]. It is produced by various types of cells, including macrophages, neutrophils, endothelial cells, and various cancer cells [36,37]. IL-8 receptors (CXCR1 and CXCR2) have been identified on many cell types, including neutrophils, T-cells, monocytes, endothelial cells, and some cancer cells [37]. Expression of IL-8 is mainly regulated by activator proteins and/or nuclear factor-kB-mediated transcriptional activity [34]. Moreover, the expression of IL-8 is regulated by a number of stimuli, including by inflammatory signals (e.g., tumour necrosis), chemical environmental stresses (e.g., exposure to chemotherapy), and steroid hormones (e.g., androgens) [34]. Studies have suggested that IL-8 is also involved in and helps with cancer progression [36,38]. Control of IL-8 expression may be a valuable tool in the design of new therapies targeting the control of cancer growth and metastasis [36]. In the case of myeloma, it has been shown that endogenous CD28 expressed on myeloma cells increases the production of IL-8 in MM patients. Moreover, abnormal CD28 progression on myeloma cells correlates with metastasis, suggesting that IL-8 plays a role in promoting myeloma metastasis [39]. The studies presented by Pellegrino et al. [40] showed that IL-8 may induce the proliferation and chemotaxis of MM cell lines and patient plasma cells. Moreover, IL-8, through its ability to enhance angiogenesis, is involved in tumour progression [37]. It has also been observed that IL-8 is involved and plays an important role in the disturbance of bone homeostasis in various types of cancer, while anti-IL-8 may be used as a potential target for the prevention of MM-induced osteolysis [37]. 

IL-8 is a promising marker for many clinical conditions and is used for rapid diagnosis or as a predictor of prognosis. Unfortunately, no studies using IL-8 as a biomarker for prognosing MM patients have been reported so far.

#### 2.1.5. IL-12

IL-12 is a cytokine that plays an important role in the interaction between the innate and adaptive arms of the immune system [41]. IL-12 is produced mainly by dendritic cells and macrophages. In smaller amounts, it can also be secreted by keratinocytes, granulocytes, and mast cells [42]. The production of IL-12 stimulates, first of all, the components of bacterial cell walls [43]. Biologically active IL-12 is a heterodimer consisting of two different subunits: p35 and p40. The p40 subunits can form a homodimer with IL-12 antagonist properties [44,45,46]. IL-12 stimulates the proliferation, activation, and cytotoxicity of T-lymphocytes and NK cells, as well as the production of IFN-γ and TNF-α by these cells [41]. IL-12 also induces the formation of Th1 lymphocytes and enhances their activity and proliferation. In addition, it stimulates the secretion of certain classes of IgG antibodies and inhibits the secretion of IgE [41]. Inhibition of vascular formation is—in addition to the stimulation of cytotoxicity of T-lymphocytes and NK cells, as well as the activation of macrophages and other food cells—the basic mechanism of the anti-tumour activity of IL-12. Due to these properties, attempts have been made to use it in cancer therapy [42]. We found no studies describing the use of IL-12 as a prognostic biomarker in MM patients.

#### 2.1.6. IL-15

IL-15 is a pro-inflammatory cytokine that is produced by macrophages, stromal cells, and fibroblasts. Its biological activity is similar to that of IL-2, especially in relation to T- and NK cells [12]. Its action supports the cellular immune response and stimulates the growth of CD4+ T-cells and induces anti-apoptotic signalling to effector CD8 + T-cells. Additionally, IL-15 promotes the proliferation and differentiation of pre-activated and normal B-cells [47]. Studies have shown [47] that stromal IL-15 production influences the growth of myeloma cells independently of IL-6, confirming the role of this cytokine in MM. Studies have been carried out [48] which showed a significantly increased concentration of this cytokine in the serum of MM patients, compared to the control. Levels of IL-15 in stage III MM patients were increased, in comparison with those in stages I and II; however, the observed difference was not statistically significant. Moreover, a correlation was recorded between the concentration of IL-15 and IL-10 in blood serum [12]. Thus, the serum level of IL-15 in MM patients is likely to be of prognostic value, but more studies should be done in the future to clearly demonstrate this.

#### 2.1.7. IL-17

IL-17 is a pro-inflammatory cytokine which is mainly secreted by memory T-cells. Its activity is mainly based on stimulating macrophages, fibroblasts, endothelial cells, and epithelium to release other cytokines [12]. The role of IL-17 in the pathogenesis of MM has been described. IL-17A has been shown to promote the growth of MM cells, but also to inhibit immune functions in the tumour environment [49], which underlines the important role of this cytokine in tumour growth and maintenance. Thus, high levels of IL-17 in patients indicate a poor prognosis for the patient’s survival [23,50,51,52]. This has been confirmed by studies [23,50,52], which showed that the levels of IL-17 in MM patients were higher than in the control group. In one study [50], IL-17 levels in stage I and II patients showed no significant differences. In the case of patients who were undergoing treatment and refractory to treatment, a correlation was observed between high levels of IL-17 and a negative response to therapy. Moreover, a correlation has been demonstrated [23] between high levels of IL-17A, IL-6, and IL-10, which are also prognostic for MM, and a negative response. Other studies have confirmed [51] that the level of IL-17 is correlated with the severity of the disease, as it was much higher in patients in stage III than in patients in stage II. Another study found [53] that a lower IL-27:IL-17 ratio in newly diagnosed MM was correlated with disease progression, while a higher value was associated with better PFS. These data indicate that determining the level of IL-17 in MM patients may be an effective factor in determining the stage of the disease and the effectiveness of the applied therapy.

#### 2.1.8. IL-18

IL-18 is a pro-inflammatory cytokine that belongs to the family of IL-1 cytokines and can be released by a variety of cells. IL-18 stimulates the killing mechanism by lymphocytes, but also influences angiogenesis, immune modulation, and bone metabolism. Depending on the circumstances, this cytokine may induce Th1 and Th2 cell reactions [12,54]. IL-18 has been shown to be strongly involved in MM, where its levels are associated with MDSC [55]. Deriving from the MM cell niche, IL-18 drives the formation of MDSC cells, which significantly accelerates the disease progression [55]. Due to the strong involvement of this cytokine in MM, studies have been conducted to verify the usefulness of IL-18 in predicting the course of the disease. Thus, an increased level of IL-18 has been demonstrated in MM patients, compared to the control group [56]. Moreover, high levels of IL-18 correlated with worse survival of patients and the severity of the disease, as these values were significantly higher in stage III patients than in stage I and II patients [57]. It is also important that high IL-18 values also correlated with increased concentrations of angiogenic cytokines [57]. These data suggest that measuring IL-18 levels in MM patients is of great prognostic value.

#### 2.1.9. IL-32

IL-32 is a pro-inflammatory cytokine, also called natural killer 4 (NK-4). Its action is related to the increased secretion of other pro-inflammatory cytokines, including IL-6 and TNF-α [58]. Studies have shown [2] that malignant plasma cells are the major source of IL-32 in MM. Moreover, paracrine-secreted IL-32α induces IL-6 production in BMSC and creates a feedback loop that promotes the growth of MM cells. The significant role of IL-32 in promoting macrophage immunosuppression in MM has also been confirmed [59]. Studies have shown [60] that, in addition to MM cells, IL-32 is also produced by the extracellular vesicles (EVs) involved in tumour growth and maintenance, which occur under hypoxic stress conditions, which characterizes the tumour microenvironment and promotes tumour progression. An analysis of IL-32 concentration showed [58] that this value was significantly higher in patients with MM than in the control group, both in the bone marrow and in peripheral blood. This has also been confirmed by another study [60], showing the high expression of the IL-32 gene in plasma cells, which was much higher in MM patients and correlated with worse survival and more advanced clinical stage of the disease [59]. Another analysis showed [61] that patients with lower IL-32 levels had more positive PFS and OS. In addition, the correlation between minimal residual disease (MRD) and IL-32 has been analysed in patients, as MRD is important in predicting disease recurrence and evaluating therapy. It has been shown [61] that MRD and high levels of IL-32 have great prognostic utility for patients, possibly enabling the early identification of resistant MM cells in patients with complete disease remission.

#### 2.1.10. Tumour Necrosis Factor Family

The tumour necrosis factor (TNF) family is a group of cytokines that includes a number of proteins, including TNF-α, B-cell activating factor belonging to TNF family (BAFF), and receptor activator of NF-κB (RANK) [24]. The TNF family is largely involved in the growth and survival of MM cells, as well as their migration and the development of drug resistance by activating specific signalling pathways [24,62,63,64]. TNF-α is a highly pleiotypic cytokine, which is involved in a wide variety of processes that control inflammation and the anti-tumour response [25]. Studies [63,65,66] have emphasized the essential role of TNF-α in tumour growth and angiogenesis in MM. It has been reported [63] that TNF-α stimulates the production of autocrine IL-6 [67], influences the expression of elements determining the avoidance of apoptosis [68], and also affects the development of drug resistance [64]. Studies have been carried out [62,69,70] which showed high levels of TNF-α cytokine in MM patients, which correlated with the severity of the disease and indicated the occurrence of severe symptoms under maintenance treatment [70]. BAFF is a cytokine necessary for the proper development and survival of B-lymphocytes, which is secreted by some T-cells, monocytes, and DC cells [12]. BAFF has been shown to promote the growth of tumour MM by an autocrine loop [71]. Studies have shown [24,72,73] increased BAFF levels in MM patients, which correlated with decreased survival and, therefore, is considered to be of prognostic value. Moreover, higher BAFF levels have been observed in relapsed or refractory multiple myeloma (RRMM) patients, compared to newly diagnosed MM, and its levels were significantly decreased after successful treatment [24,74,75]. RANK is a cytokine mainly produced by osteoblastic lineage cells and stromal cells, which is involved in cell growth and differentiation [76]. Literature data have indicated [77] that RANK signalling and its ligand RANKL are involved in tumour formation and growth, and that tumour cells or osteoprogenitor cells may be responsible for RANKL release. The conducted studies have suggested [78] that RANKL is gradually upregulated as MM progresses, where its inhibition results in a delay in disease progression and related bone damage, as observed in a mouse model. High levels of RANKL in the blood serum of MM patients have been demonstrated [24,79], with the level of this marker reflecting disease severity, lytic bone damage, and poor prognosis for MM patients. Other studies have shown [80,81] that a high ratio of RANKL to its endogenous antagonist osteoprotegerin (RANKL:OPG) characterized MM patients with shorter survival, where high levels of soluble RANKL correlated with the degree of bone damage [80,81].

#### 2.1.11. Interferon-𝛾 (IFN-𝛾)

It has been shown that IFN-𝛾 inhibits myeloma cell proliferation, on the basis of the IL-6-dependent myeloma cell lines and fresh bone marrow cells [82]. It has been concluded that the antiproliferative effect of IFN-𝛾 is dependent on IL-6 inhibition, being the central myeloma growth factor [82]. The inhibitory IFN-𝛾 effect may be compared with the effect achieved by the use of dexamethasone, while the combination of IFN-𝛾 and IFN-𝛼 resulted in an inhibitory effect on proliferation and immunoglobulin synthesis [82]. On the other hand, studies exist which claim that no beneficiary effect was seen by the use of IFN in treating MM patients [83]; further, to reap the benefits from the IFN-𝛾 immunomodulating feature, a schedule of IFN-𝛾 therapy should be implemented [84]. IFN-𝛾 can also upregulate the MHC class II transactivator (CIITA), which leads to MHC class II expression, therefore leading to the rapid recognition of stressed tissues [85,86]. Additionally, IFN-𝛾 acts on tumour cells, enhancing their recognition by CD8+ and CD4+ T-cells, thus further activating macrophages in the tumour, leading to tumour growth inhibition [87].

As far as MM treatment is concerned, it has been shown that IFN-𝛾 induces CD20 expression on multiple myeloma bone marrow plasma cell (BMPC) and B-cells, which may be a facilitating factor for the use of rituximab, which binds to MM BMPCs and may serve as the direction of serotherapy in the chosen group of patients [88].

### 2.2. Anti-Inflammatory Cytokines

#### 2.2.1. IL-1Ra

IL-1Ra is an anti-inflammatory cytokine that regulates IL-1, as it binds to the receptor without activating it. IL-1Ra acts as a neutralizer for IL-1 and, thus, inhibits the inflammatory and pro-tumour properties of this cytokine, reducing the activity of IL-1 by up to 95% [89,90]. Three variants within IL-1Ra have been described: sIL-1Ra, which is secreted from monocytes, macrophages, neutrophils, and other cells; icIL-1Ra, which remains in the cytoplasm of keratinocytes and other epithelial cells, monocytes, and fibroblasts; and a form of IL-1Ra, which has been described in neutrophils, monocytes, and liver cells [91]. Studies have demonstrated the production of IL-1Ra in myeloma cell lines, as well as in the peripheral blood and bone marrow of MM patients. Moreover, IL-1Ra values in MM patients are elevated in the bone marrow [92]. A study of cytokine levels in MM patients treated with bortezomib showed that serum IL-1Ra levels were higher in stage III patients than in stage I/II patients. Moreover, high levels of IL-1Ra have been associated with bone involvement in MM [93]. Importantly, the use of IL-1Ra has significant value in anti-IL-1 therapies, as it has been described to effectively inhibit IL-1-induced paracrine IL-6 production following the use of IL-1Ra [94]. In a clinical trial (NCT00635154), the use of IL-1Ra treatment in combination with dexamethasone in patients suffering from mild precursor conditions with the risk of developing an active form of MM caused a reduction in myeloma growth and an improvement in PFS [90] as, in addition to the maximum inhibition of IL-6 synthesis, apoptosis of myeloma cells was also observed [90]. In addition, it has been shown that, within 6 months of treatment with IL-1Ra, the level of CRP in patients was reduced, which was associated with a decrease in the proliferation of cancer cells [14]. It has also been reported [94] that increased levels of IL-17 may be useful in identifying patients for whom IL-1Ra treatment will be effective. High levels of IL-17 may indicate too advanced inflammation, thus limiting the effectiveness of this therapy. A clinical trial combining IL-1Ra therapy with dexamethasone and lenalidomide for the treatment of early stages of MM is in Phase I (NCT02492750).

#### 2.2.2. IL-4

IL-4 is a cytokine produced by Th2 cells, NKT lymphocytes, mast cells, and basophils [95]. IL-4 receptors are found on T- and B-lymphocytes, on mast cells, monocytes, macrophages, and fibroblasts, as well as on hematopoietic cells and many others. There are two types of IL-4: IL-4R type I and IL-4R type II [42]. IL-4 has a wide range of interactions. One of the most important interactions is the effect of IL-4 on the proliferation and differentiation of B cells. There is also a correlation between IL-4 secretion and plasma IgE levels, due to the effect of this cytokine on B-cell class switching [42]. Research on IL-4 has revealed a great deal of information about its various physiological and pathological roles [96]. Physiologically, IL-4 acts directly on tumour cells as a tumour-promoting cytokine [97]. IL-4 has been observed to have a significant effect on tumour progression. Increased production of IL-4 has been confirmed in cancer of the prostate, breast, lung, and kidney cells, as well as many other types of cancer [96]. Unfortunately, we did not find any studies proving the prognostic importance of IL-4 in patients with MM.

#### 2.2.3. IL-10

IL-10 is an anti-inflammatory cytokine which is produced by monocytes, macrophages, NK cells, T- and B-lymphocytes, and mast cells [12]. Its action is based on the inhibition of the immune response and acts on the mechanisms of innate and adaptive immunity [12]. For this reason, IL-10 may halt the release of pro-inflammatory cytokines, antigen presentation, and cell growth [98]. IL-10 has been reported to be of great importance in tumour growth and maintenance. Apart from the fact that it likely promotes the escape of cancer cells from the immune system, IL-10 significantly influences the growth of B-cells and their differentiation into plasma cells, as well as the production of immunoglobulins [99]. Thus, it has been described that abnormal levels of IL-10 released by CD8 + T-cells and MM cells may support the MM immunosuppressive environment by abolishing the function of DCs [100]. IL-10 acts as a proliferative factor for plasma cells, but also supports angiogenesis in MM [101]. Due to the fact that IL-10 has a documented importance in MM, studies have been conducted to analyse the relationship of this cytokine level with the course of the disease. Analysis of the results confirmed the increased amounts of IL-10 in the blood serum of MM patients, compared to the control group [48,99,101,102]. The conducted studies showed that high levels of IL-10 in the blood serum in patients with the initial stage of MM negatively influenced PFS and OS, treatment response, and prognosis [102]. Other independently conducted studies [48,99] also confirmed that high levels of IL-10 were associated with the severity of the disease in MM patients. These data clearly indicate that the determination of IL-10 in the blood serum of patients is a non-invasive prognostic tool for predicting the stage of MM and clinical management.

#### 2.2.4. IL-22

IL-22 is an anti-inflammatory cytokine that belongs to the IL-10 family of cytokines and is secreted by activated Th1 lymphocytes, endothelial cells, NK cells, activated dendritic cells, and histiocytes. IL-22 affects the innate immune system and controls the acute phase response, as well as cell differentiation and migration [12]. It has been shown [103] that an increased amount of IL-22 can stimulate MM growth and influence the development of immunosuppression in the tumour environment. A study has shown [103] significantly higher levels of IL-22 in patients with active form of MM, compared to healthy people and patients with disease remission, as well as higher levels in patients with disease remission, compared to the healthy group. Additionally, high levels of IL-22 have been correlated with the severity of the disease. It has also been recorded [103] that, with increasing IL-22 levels, increasing IL-1ß concentration was also observed. These data indicate a high potential for the use of IL-22 as a prognostic factor for MM; however, more studies should be carried out to clearly confirm this.

### 2.3. Growth Factors

The roles of several growth factors in the course of multiple myeloma have been confirmed. As the process of angiogenesis is of high significance in the progression of MM, the imbalance between antiangiogenic and proangiogenic factors may be the reason for angiogenic switch, leading to the vascular phase of the disease [104,105]. It has been proven [106] that the elevated levels of growth factors give rise to the use of antiangiogenic drugs in curing MM; however, among those particles, some seem to be good candidates for prognostic factors in this disease. Here, one may enumerate FGF-2 (Fibroblast growth factor 2), VEGF (Vascular endothelial growth factor), HGF (Hepatocyte growth factor), PDGF-β (Platelet-derived growth factor), and ANG-2 (Angiopoietin -1). 

Studies have been performed in which statistically significant decreases in the levels of FGF-2 and VEGF have been registered in MM patients [107]. A low level of FGF-2 has been associated with a shorter time to MM progression, as investigated through ROC [107], where those results were consistent with previous studies based on single samples per participant [108]. These results led to the conclusion that low blood levels of VEGF and FGF-2 might serve as a prognostic factor in MM, as these growth factors are associated with tumour angiogenesis [109]. Interestingly, such a use of growth factors, serving as prognostic factors on the basis of the negative correlation of the levels of FGF-2 and VEGF, has also been confirmed in a study involving pre-clinical blood samples from patients many years before clinical MM diagnosis [108]. Moreover, VEGF has been positively correlated with IL-20 levels in patients with MM, suggesting that the link between these two parameters may be used as an indicator of the disease progression and angiogenesis process [110]. In another study [111], it was confirmed that the contribution to the development of MM may lie not only in the elevated levels of VEGF and FGF, but also the polymorphism of the genes coding these growth factors. In this research, it was suggested that the bFGF G allele, in particular, may be a potential marker associated with worse response to therapy in MM patients [111]. Clinical trials with VEGF impact on MM patients have also been performed. In one of the trials (NCT03136653), a drug called MP0250—which is able to neutralize the activities of VEGF and HGF and simultaneously has the ability to bind to human serum albumin (HSA)—was examined. Although, in theory, MP0250 should increase plasma half-life and the tumour penetration, the results of this clinical study have not been posted [112]. Similarly, no results are known from the clinical trial (NCT00047788) involving another drug impacting the potential of VEGF in MM—ZD6474 [113]. Another clinical trial (NCT00410605) involving the role of VEGF in MM patients receiving bevacizumab with lenalidomide and dexamethasone was aimed at determining the overall response rate [114]. Blood samples were checked for the VEGF and VEGFR (VEGF receptor) polymorphism. Another clinical trial (NCT01183663) involving anti-VEGF monoclonal antibody (bevacizumab) was aimed at defining the highest tolerable doses of the safe combination of drugs effective in advanced cancer treatment [115]; nevertheless, no significant result has been achieved for MM patients [116]. Furthermore, a comparison of serum angiogenic markers Ang2, G-CSF, follistatin, HGF, FGF-1, endothelin 1, and VEGF-A has been carried out between Monoclonal gammopathy of undetermined significance (MGUS) and smouldering myeloma (SMM/MM) groups of patients (clinical trial NCT01237054) [117]. The results of this clinical study showed that VEGF levels were elevated in MM/SMM patients, in comparison to MGUS [118,119].

Another growth factor with potential therapeutic impact in MM is HGF (hepatocyte growth factor). As the receptor of HGF, Met, is expressed on MM cells, HGF may be a candidate for changing the picture of the disease. It has been shown that stimulation of MM cells with HGF leads to the activation of signalling pathways such as PI3K/PKB (phosphatidylinositol 3-kinase/protein kinase B) and RAS/mitogen-activated protein kinase, indicating that it can be implicated in the regulation of cell proliferation and survival [120]. The enhanced proliferative and anti-apoptotic effects of HGF on MM cells have also been noted [120]. Moreover, there exist studies in which a high level of HGF was associated with an unfavourable prognosis [121]. HGF levels were significantly lower at the time of response than at diagnosis, with an increase noted only in a low percentage of patients [122]. In the study of Seidel et al. [122], it was concluded that HGF may serve as a useful factor for predicting the outcome of the disease, as HGF levels in the bone marrow of the patients with MM are usually elevated. A median concentration of HGF concentration in bone marrow has been estimated for 10 ng/mL, a ratio of 1:10 in the mean concentration seen in blood serum, which is 1 ng/mL [122].

This has also been confirmed by the study of Kara et al. [123], who showed that there exists a symbiotic relationship between the growth of myeloma cells and HGF in patients with MM. In summary, it may be concluded that many patients with MM have increased levels of HGF; however, whether this is a result of HGF production by the malignant cells remains questionable [122]. Nevertheless, high serum HFG is unfavourable for the patient, with an impact on the survival and the response to treatment [122]. 

Another growth factor with a confirmed role in MM worth mentioning is PDGF-β (platelet-derived growth factor β) [123,124]. It has been reported that PDGF-β may be a potential stimulator of angiogenesis in many types of solid and hematological malignancies, as confirmed in the study of Tsirakis et al. [124], who registered a positive correlation between serum PDGF-β and microvascular density (MVD) in MM patients, meaning that patients with higher MVD were characterized by higher serum levels of PDGF-β. Moreover, a significant difference was noticed in serum levels of PDGF-β in patients before and after treatment, where the survival time was higher in patients with low MVD, in comparison to the patients with high levels of PDGF-β. Studies have confirmed that PDGF-β may be an important indicator of the MM patient’s immunological status, but is of greater value in the advanced stage of the disease [123].

A factor strictly interacting with microvascular density is angipoietin-1 (Ang-2), the role of which has also been depicted in several malignancies of different origin [125]. With the use of Western blot, RT-PCR, and immunocytochemistry it has been proved that the major source of Ang-2 in MM patients are endothelial cells, impacting the activation of those cells [125]. Data suggest that Ang-2 produced in bone marrow may contribute to MM angiogenesis and may also serve as a reliable angiogenesis biomarker [125]. On the other hand, Ang-1 is expressed by several human myeloma cell lines (HMCLs) at the mRNA and protein level, influence vessel formation, and may also be partly involved in MM angiogenesis [126]. Moreover, the study of Terpos et al. [127] showed that the Ang-1/Ang-2 ratio may also be useful in MM.

Another growth factor worth mentioning, which is known to serve as a proliferative and anti-apoptotic factor in MM, is IGF-1 (insulin-like growth factor 1) [128,129,130]. Interestingly, on the other hand, evidence exists that IGF-1 promotes migration by acting as a chemoattractant and impacts invasion in MM [131,132,133]. The migratory properties of IGF-1 are upregulated and correlated with epithelial–mesenchymal transition (EMT) markers in MM patients, which provides proof of the potential use of this growth factor in MM treatment [134]. It has been confirmed that IGF-1 induces activation of NF-κB in MM cells and reduces the Apo2L/TRAIL sensitivity of these cells [128]. Interestingly, IGF-1 is mainly produced by the bone microenvironment, but is also present in the peripheral blood serum of MM patients at significantly increased levels [128]. The anti-apoptotic effect obtained by the IGF-1 differs from that achieved by IL-6 in MM individuals [128].

### 2.4. Chemokines

#### 2.4.1. MIP-1 (Macrophage Inflammatory Protein-1, CCL3) and CCL2 (MCP1, Monocyte Chemoattractant Protein 1)

MIP-1 is a member of the CC chemokine family, which are primarily associated with cell adhesion and migration. Nevertheless, the role of this chemokine in the course of multiple myeloma has been reported several years ago. The level of MIP-1 is increased in patients with MM, especially in bone marrow plasma, and it has been noted that the levels of this protein may be correlated with the activity and stage of the disease [135]. Unfortunately, high levels of serum MIP-1alpha have been associated with unfavourable diagnosis [135]. On the other hand, it was also registered that MIP-1alpha may be indirectly associated with survival: The 3-year probability of survival was 85% and 44% for MIP-1alpha levels below and above 48 pg/ml, respectively (*P* = 0.021) [136].

A positive correlation exists between MIP-1aplha and beta-2-microglobulin, meaning that this chemokine—apart from having a strong osteoclast activity—may be implicated in myeloma growth and survival [135]. This was also confirmed by the study of Abe et al. [137], who registered that MIP-1alpha enhances osteoclastic bone resorption in MM patients, being a major osteoclast-activating factor produced by MM cells. The disruption of MIP-1 with its cognate receptor may be critical for targeting the treatment, based on osteolysis [138].

The MIP-1α signalling receptors—CCR1 and CCR5—are expressed by human myeloma cell lines [139]. Moreover, 5TGM1 cells have been shown to express mouse CCR1 receptors, which led to the conclusion that MIP-1α may also have a potential autocrine effect on myeloma cell growth and survival [139]. Roodman and Choi [140] claimed that, when looking at the role of MIP-1 in MM, neutralizing its receptor may provide new treatment alternatives for both tumour action and bone destruction.

#### 2.4.2. CCL2 (MCP1, Monocyte Chemoattractant Protein 1) and CCL3

Overexpressing chemokines, such as CCL2 (MCP-1) and CCL3, accompanying the recruitment of macrophages, is a common phenomenon in human tumours [40,141]. In MM, the mechanism explaining the increased number of tumour-associated macrophages (TAM) in unclear; however, the roles of CCL2 and CCL3 in the process have been confirmed [40,141]. One of the impacts of CCL2 and CCL3 in MM may be the regulation of osteoclast-mediated bone resorption [135]. The results of Li et al. [142] showed that CCL2 and CCL3 are functional chemokines, responsible for increased infiltration of macrophages in the MM bone marrow environment and elevated polarization into TAM. Recently, further details have been added to this polarization, on the basis of the study of Xu et al. [143], who showed that this polarization towards M2-like phenotype macrophages helps to protect MM cells from drug-induced apoptosis. This may be useful for the improvement of chemotherapy, which is often blocked by chemoresistance [142,143].

A summary of the roles of cytokines and chemokines in MM is shown in Figure 1. The prognostic values of the selected immunological factors impacting the treatment of MM patients are summarized in Table 1.

## 3. Immune System Cells

### 3.1. Neutrophil to Lymphocyte Ratio (NLR), Monocyte to Lymphocyte Ratio (MLR), and Platelet to Lymphocyte Ratio (PLR)

Systemic inflammation is one of the hallmarks of cancer, and the body’s tumour-related inflammatory response plays one of the key roles in enhancing oncogenesis [146]. Systemic inflammation is associated with changes in peripheral blood leukocytes. These changes can be captured by the use of cheap and easy to determine biomarkers, in the form of Neutrophil to Lymphocyte ratio (NLR), Monocyte to Lymphocyte ratio (MLR), and Platelet to Lymphocyte ratio (PLR). These biomarkers have aroused interest in the scientific community for many years. Thus far, their influence on clinical outcomes has been demonstrated in many solid tumours [146,147,148,149,150,151,152,153,154,155,156]. 

Neutrophil to Lymphocyte ratio (NLR) is calculated by dividing the neutrophil count by the lymphocyte count. Elevated NLR levels have been shown to be largely caused by neutrophils and lymphopenia [157]. Occurring neutrophilia may stimulate the secretion of active cytokines, such as VEGF, which largely accelerates tumour progression, while lymphopenia is believed to correlate with the severity of the disease and affects the escape of cancer cells from the surveillance of the immune system [157,158,159]. To date, many studies have shown the prognostic role of NLR in many cancers, diseases, and therapies, including cancers of the colon, urinary tract epithelium, ovaries, prostate, oesophagus, pancreas, and kidneys [147,148,149,150,151,152,153,154,155]. Studies that have investigated the role of NLR in multiple myeloma showed that patients showed relatively higher levels of NLR, as compared to the control group (Table 1). It was also noted that the increased level of NLR was able to predict the shortening of OS and PFS in patients with MM [156].

The Monocyte to Lymphocyte ratio (MLR) is another, next to the NLR, inexpensive, easy to measure, and reproducible marker used to assess systemic inflammation [160]. It is calculated by dividing the number of monocytes by the number of lymphocytes [161,162,163]. To date, MLR has been proven as a prognostic factor in cancer and tuberculosis patients, and may also serve as an independent risk factor for CVD and a predictor of the severity of the lesion in patients with previous coronary artery disease [160,161,162,163]. Studies that have assessed the contribution of MLR in patients with MM showed that, as in the case of NLR, a relatively higher level of MLR is observed in MM patients, compared to the control group. It was shown that a high MLR level, as in the case of NLR, was able to predict the shortening of OS and PFS in patients with MM [156,164]. 

The Platelet to Lymphocyte ratio (PLR) is a relatively new inflammatory marker that, like NLR and MLR, can be used to predict systemic inflammation and mortality in many diseases, including various malignancies [165,166]. PLR is calculated by dividing the platelet count by the lymphocyte count [146,167,168,169,170,171]. Studies that have investigated the contribution of PLR to mortality prediction in patients with haematological and non-haematological malignancies have shown reactive thrombosis to be associated with inflammatory responses, while many studies have investigated the correlation between PLR and patient prognosis, with high PLR being reported as a predictor of poor prognosis [146,168,169,170,171]. Contrary to previous studies, it has been shown that, in patients with multiple myeloma, a low platelet count or low PLR level is considered a poor prognostic factor. The difference in MM may be due to the pathology of the disease [156,166]. 

Summarizing the previous findings, it can be concluded that elevated NLR and MLR levels and decreased PLR levels are associated with adverse clinical and biological features and shortened OS and PFS in MM patients (Table 1). Therefore, it can be considered reasonable to use NLR, MLR, and PLR as biomarkers to estimate clinical status and survival in MM patients.

### 3.2. T-cell Subpopulations

CD4 + T-cell subpopulations, such as Treg and Th17 cells, have been shown to be crucial in cancer resistance. Treg cells constitute 5–10% of all CD4 + T cells in peripheral blood [172]. Tregs are largely associated with impaired immune function in both solid tumours and haematological neoplasms [173,174], while the role of Th17 cells in the pathogenesis of cancer is still not well-described or -defined [175]. However, the balance between Treg and Th1 cells is very important in maintaining proper homeostasis of antitumour immunity [175]. FoxP3 is believed to be one of the major regulatory molecules in Treg cells. This molecule is expressed in natural Treg cells and Treg CD4 + CD25hi + FoxP3 cells. Treg, through contact-dependent and -independent mechanisms, suppresses other immune cells [176,177,178]. 

The research presented by Muthu Raja et al. [179], in which Treg cells were tested in patients with MM, showed an increased incidence of Treg in patients with newly diagnosed and relapsed multiple myeloma, compared to the control group. In addition, Naïve and activated Treg cells were found to be significantly greater in MM patients than in controls [179]. Functional studies have shown that Treg and their subgroups in both MM patients and controls showed similarity in inhibitory function. A significantly increased frequency of Treg was found in MM patients with unfavourable clinical symptoms [179]. Braga et al. [180] showed that, in patients with MM, the expression of FoxP3 was six times higher and that of CTLA4 was thirty times higher than in the control group. Median overall survival in patients with MM was 16.8 months [180]. One-way analysis showed that none of the genes associated with CD4 + T-cells had an impact on patient prognosis [180].

In conclusion, the knowledge so far about the involvement of Treg and Th17 cells in the progression of multiple myeloma has not been fully elucidated and, as such, more research is needed on the participation of Treg and Th17 in the progression and prognosis of MM patients.

### 3.3. Circulating Plasma Cells 

Testing with flow cytometry to detect Circulating Plasma Cells (PCs) is considered one of the predictors of survival in patients with MM. Few studies have been published examining the relationships of circulating PCs in patients with MM [181].

In 2005 [181], a study was published which showed that MM patients with 10 or fewer circulating PCs had higher overall survival than patients with more than 10 circulating PCs. In a multivariate analysis, the authors noted that the prognostic value of circulating PCs was independent of β 2-microglobulin, albumin, and C-reactive protein; moreover, there was a weak correlation between tumour weight and circulating PCs, which may suggest that the appearance of circulating PCs may be reflected in the tumour biology [181]. The authors concluded that the measurement, by flow cytometry, of the number of circulating PCs in patients with newly diagnosed MM is an independent prognostic factor for survival [181].

However, due to the fact that the prognostic values of these factors are not easily reproducible, it has been concluded that these factors are not a very good predictor and are currently not widely accepted [182].

## 4. Other Markers

### 4.1. Junctional Adhesion Molecule-A (JAM-A)

Junctional adhesion molecule-A (JAM-A) (or JAM-1, CD321, F11R) is a transmembrane protein belonging to the immunoglobulin superfamily, encoded by the *F11R* gene [183,184,185,186]. JAM-A is expressed at cell–cell junctions and the protein has been shown to have a complex set of functions in epithelial and endothelial cells [184,185]. In these cells, the JAM-A protein influences the regulation of the epithelial barrier, the migration of endothelial cells, the angiogenesis process, the aggregation of thrombocytes, and the leukocyte adhesion process [185,186]. Aberrant JAM-A expression or deregulation has been observed to confer a much more aggressive phenotype in people with poor prognosis for various types of cancer, including lung, breast, brain, head, and neck cancer, as well as multiple myeloma [186,187,188]. Excessive JAM-A activation results from either elevated levels of abnormal dimerization, leading to constitutive receptor signalling, or from excessive release into the microenvironment of JAM-A ligands by normal and neoplastic cells [189]. Membrane-bound JAM-A and its soluble form (sJAM-A) can form homophilic, as well as heterophilic, interactions. Membrane-bound JAM-A and its soluble form (sJAM-A) can form homophilic and heterophilic interactions with LFA-1 (lymphocyte function-associated antigen 1), AFDN (afadin), CASK (calcium/calmodulin-dependent serine protein kinase), and TJP1 (tight junction protein-1) with high receptor/ligand binding affinity. These interactions influence the activation of the JAM-A signalling pathway, which is involved in the regulation of survival, growth, angiogenesis, and metastasis of cancer cell spread [186]. 

To date, few studies have been published evaluating the effects of JAM-A in multiple myeloma. In one of the studies [190], it was shown that, in plasma cells collected from MM patients, elevated levels of sJAM-A were correlated with poor prognosis. The authors also showed that sJAM-A levels were significantly elevated in patients with MM, relative to controls [190]. In vitro JAM-A inhibition studies confirmed the effects of JAM-A on the impairment of MM cell migration, colony formation, chemotaxis, proliferation, and viability. Furthermore, we observed that treatment (in vivo), with an anti-JAM-A monoclonal antibody (αJAM-A moAb) had an inhibitory effect on tumour progression in a mouse xenograft MM model [190]. In contrast, another study [186], which evaluated whether MM BM endothelial cells (MMECs) control disease progression through JAM-A, showed that membrane and cytoplasmic JAM-A levels were significantly elevated in MMECs from NDMM patients and RRMM patients, compared with MGUS and controls. In MMECs, elevated levels of membrane JAM-A expression predicted poor clinical outcome [186]. The addition of recombinant JAM-A to MMECs has been shown to enhance angiogenesis, while its inhibition impaired angiogenesis and MM cell growth in 2D and 3D in vitro cell cultures and membrane assays. Studies have shown that elevated JAM-A expression in bone marrow endothelial cells may be an independent prognostic factor for the survival of NDMM and RRMM patients [186]. 

In conclusion, JAM-A is a promising diagnostic and therapeutic target, which can be used as a biomarker in MM patients.

### 4.2. Autophagic Markers: Beclin-1, LC3

Autophagy is a process of intracellular self-degradation, which balances the cellular energy sources and regulates tissue homeostasis [191]. Autophagy, in the physiological state, directs cytoplasmic components to autophagolysosomes for degradation and is one of the alternative behaviours that causes cell death [191]. The process of autophagy is a very complex process, regulated by various multistage signalling pathways [192]. Under basal conditions, it is slightly activated in a large number of human cells, in order to maintain physiological homeostasis and to promote cell survival under stress conditions. Any mutations occurring within the autophagy gene and/or within changes in the regulatory mechanism result in the onset or progression of various human disorders and diseases, including cancer [193]. To date, several key markers involved in autophagy have been identified. Two such markers are Beclin-1 and LC3 [194]. Beclin-1 and LC3 are involved in the initial nucleation of the insulating membrane and the subsequent extension step. The development of IHC for autophagy-related molecules made it possible to study their prognostic significance in various cancers [195]. In studies [196] that assessed the effect of Beclin-1 and LC3 immunoreactivity in patients with confirmed multiple myeloma, it was shown that patients with high levels of Beclin-1 and LC3 immunoreactivity had significantly better overall survival, compared to patients with moderate or low Beclin-1 or LC3 immunoreactivity [196]. Additionally, the authors showed an inverse correlation of Beclin-1 and LC3 expression with serum globulin levels, which may suggest that high levels of Beclin-1 and LC3 expression may result in higher autophagic activity; hence, there may be better survival in the Beclin-1 group and Highly expressed LC3, suggesting a possible cytoprotective effect of autophagy in multiple myeloma [196]. 

Summarizing this research, it can be concluded that higher immunoreactivity towards autophagy markers (Beclin-1 and LC3) in multiple myeloma is associated with better overall survival of MM patients.

### 4.3. Immune Checkpoint Sygnalling: PD-L1

Programmed death ligand 1 (PD-L1; also known as B7-H1 or CD274) is a type 1 transmembrane protein encoded by the PDCDL1 gene, belonging to the immunoglobulin (Ig) superfamily [197,198,199,200]. PD-L1 was the first functionally characterized ligand of the programmed death receptor 1 (PD-1) [197,199,200]. It consists of the extracellular IgV-like and IgC-like domains, as well as a hydrophobic transmembrane domain and a short cytoplasmic tail containing no canonical signalling motifs [200,201,202]. PD-L1 in tumour cells has shown intrinsic effects, including increased proliferation, invasion and migration of cells, increased drug resistance, and decreased apoptosis [203]. Expression of PD-L1 is one of the mechanisms of immunity avoidance, which is used in various types of cancer, and is associated with a worse prognosis [199]. PD-L1 expression has been observed on hematopoietic cells (T- and B-lymphocytes, macrophages, dendritic cells, and mast cells) and healthy non-hematopoietic tissue cells (vascular endothelial cells, keratinocytes, astrocytes, Langerhans islands, syncytiotrophoblast cells, corneal epithelial cells, and coronary endothelial cells) [200]. PD-L1 and its PD-1 ligand belong to the immune checkpoint pathway [197]. The PD-1/PD-L1 interaction under physiological conditions is important for the development of immune tolerance and prevents the excessive activation of immune cells, which can lead to autoimmunity and tissue destruction [199]. The PD-L1–PD-1 pathway has proven its value as a therapeutic target for many types of malignant neoplasms among all immune pathways. Currently, treatments targeting the PD-L1–PD-1 axis have been evaluated in more than 1000 clinical trials and have already been approved for some cancers, including melanoma, renal cell carcinoma, Hodgkin’s lymphoma, squamous cell carcinoma of the head and neck, bladder cancer, non-small cell lung cancer Merkel cell carcinoma, and unstable solid tumours (tall microsatellite or mating deficient solid tumours) [200]. 

In neoplastic cells, the mechanisms of PD-L1 induction are mainly associated with proinflammatory cytokines and oncogenic and transcriptional pathways, which include PTEN, mTOR, and PI3K [204]. In multiple myeloma, multiple signalling pathways have been shown to increase PD-L1 levels on myeloma cells so far. Up-induction was regulated, inter alia, by STAT1 activation, which was stimulated by IFN-γ, or TLR activation by the MyD88/TRAF6 or MEK/ERK pathway [204]. Recent studies have shown that patients with MM showed significantly higher levels of PD-L1 on myeloid and plasmacytoid dendritic cells, compared to the control group, while a significant correlation was also noted in the percentage of MM PD-L1^+^ cells and CD141^+^ myeloid dendritic cells [205]. Expression of PD-L1 on MM cells has been associated with a greater proliferation potential and resistance to anti-myeloma agents, suggesting that PD-L1 expression plays a key role in the progression of multiple myeloma [205].

The prognostic significance of PD-L1 expression in multiple myeloma is largely unknown. In studies using immunohistochemical staining to assess PD-L1 expression in bone marrow biopsy specimens of multiple myeloma patients receiving autologous stem cell transplant (ASCT) at various time points (i.e., before ASCT, after ASCT, and/or at ASCT relapse), it has been shown that sustained/achieved PD-L1 expression may be one of the unfavourable prognostic markers of overall survival after autologous stem cell transplantation [206].

The impact of the discussed parameters is summarized in Table 2.

## 5. Further Directions

The immune system and its individual components play an important role in the pathogenesis of multiple myeloma. It has been documented that the immune cells present in the tumour environment and the associated released cytokines significantly influence the progression of MM in patients. Clinical studies have confirmed that the analysis of changes in the levels of these factors has prognostic value, thanks to which it is possible to monitor the response of patients to MM and predict the further course of the disease. However, more research on the role of these components of the immune system as prognostic factors is needed, in order to unequivocally verify their reliability. This widened knowledge may be of special use in establishing novel and future directions for MM treatment, especially incorporating combination therapies involving more than one biomarker. Not only should the prognostic immunological factors be incorporated in the treatment, but also the overall potential of the immune response; for example, the next step might be engineered T-cell approaches, such as CAR-T cell therapies, which have been successfully used in several different types of carcinomas [207]. Data on the prognostic factors in MM may serve as a solid background for efficient and correlative therapy with CAR-T cells and should set the direction for further investigations.

Another important aspect for future consideration is the impact of epigenetic changes, such as DNA methylation, histone modification, and non-coding RNA deregulation [208]. One example of the combined use of several proteins, such as tumour suppressor proteins, growth factors, and mRNAs of oncogenic proteins (known as Selinexor), has already been approved in MM [209].

A crucial, but troublesome, issue for MM therapy in the future is vaccination. Since the disease causes treatment-related immune dysfunctions, this is a complicated, but ongoing, process [210]. 

With no doubt, future treatment should be patient-oriented, and as individual as possible, and should rely on the most effective combination of the available therapies.

## Figures and Tables

**Figure 1 ijms-22-03587-f001:**
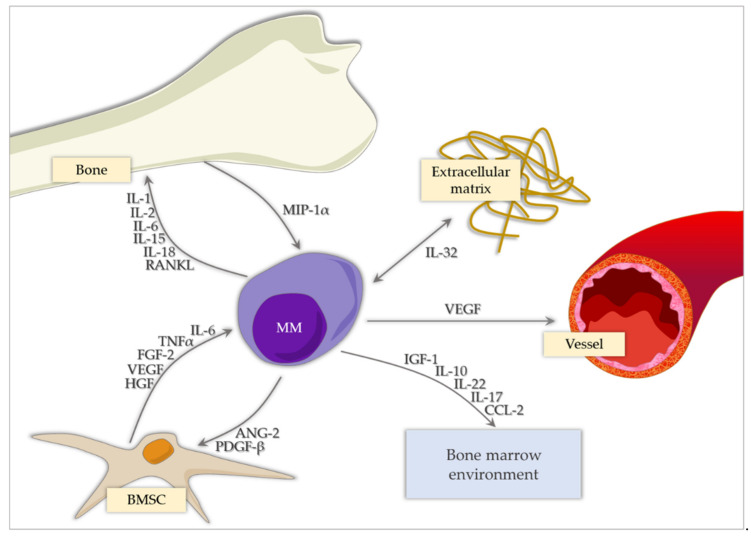
The role and impact of chosen cytokines and chemokines in the course of MM. BMSC, bone marrow stem cell.

**Table 1 ijms-22-03587-t001:** Prognostic value of selected immune system elements in multiple myeloma patients.

	Factor	Function in MM	Prognostic Value	Reference
Pro-inflammatory interleukins	IL-1	Promoting the invasiveness and progression of the tumour; Stimulation of IL-6 production; IL-1 plays a role in the conversion of latent myeloma to active MM.	High levels of IL-1 correlate with poor prognosis.	[11,12,14]
	IL-2	The IL-2–IL-2R system plays a key role in maintaining the proper functioning of T-cells.	Disturbance of the IL-2–IL-2R system may be a prognostic marker in the diagnosis of MM in comparison with MGUS; malignancy marker; high levels of sIL-2R correlate with poor PFS.	[19,20,21]
	IL-6	Promotion of tumour growth and reduction in apoptosis in myeloma cells by the JAK/STAT and RAS/MAPKs pathways.	High levels of sIL-6R are associated with shorter survival; elevated levels of IL-6 and sIL-6R reflect the level of disease activity and indicate poor prognosis; high cellular expression of IL-6 mRNA in MGUS patients may predict the development of MM.	[12,22,26,33]
	IL-15	The production of IL-15 in the stroma influences the growth of myeloma cells independent of IL-6; overexpression of IL-15 in MM plasma cells protect them against apoptosis.	The levels of IL-15 in stage III MM patients are increased, compared to stage I and II patients.	[12,48]
	IL-17	IL-17A promotes the growth of MM cells and inhibit immune functions in the tumour environment.	High levels of IL-17 indicate poor prognosis, negative response to therapy, and correlate with disease severity; IL-27: IL-17 ratio in newly diagnosed MM correlates with disease progression.	[23,49,50,51,52,53]
	IL-18	IL-18 influences the formation of MDSC cells and increased levels of angiogenic cytokines.	High levels of IL-18 correlate with poorer patient survival and the severity of the disease.	[55,56,57]
	IL-32	IL-32α induces IL-6 production in BMSC and creates a feedback loop that promotes MM cells growth; IL-32 promotes macrophage immunosuppression.	High expression of the IL-32 gene in plasma cells correlates with worse survival and more advanced clinical stage of MM; lower IL-32 levels result in more positive PFS and OS; high levels of IL-32 likely enable the early identification of resistant MM cells in patients with complete disease remission.	[59,60,61]
Anti-inflammatory interleukins	IL-1Ra	Regulation of IL-1 activity.	Serum IL-1Ra levels were higher in stage III patients than in stage I/II patients after bortezomib therapy; high levels of IL-1Ra are associated with bone involvement in MM.	[1,2,3]
	IL-10	Abnormal levels of IL-10 released by CD8+ T-cells and MM cells may support the MM immunosuppressive environment by abolishing DC function; IL-10 acts as a proliferative factor for plasma cells, but also promotes angiogenesis in MM.	Elevated serum IL-10 levels in patients with initial MM negatively affected PFS and OS, response to treatment, and prognosis, but is also associated with disease severity in patients with MM.	[48,99,100,101,102]
	IL-22	IL-22 can stimulate the growth of MM and influence the development of immunosuppression in the tumour environment.	High levels of IL-22 correlate with the severity of the disease.	[103]
Tumour necrosis factor family	TNF-α	TNF-α stimulates the production of autocrine IL-6.	High levels of TNF-α in patients with MM correlate with the severity of the disease and indicated the occurrence of severe symptoms during maintenance treatment.	[67,68]
	BAFF	BAFF promotes the growth of tumour MM by an autocrine loop.	Increased BAFF levels in patients with MM correlate with decreased survival.	[24,71,72,73]
	RANK	RANK signalling and its RANKL ligand are involved in tumour formation and growth.	Level of RANK reflecting disease severity, lytic bone damage and poor prognosis for MM patients; high RANKL: OPG ratio characterize MM patients with shorter survival, and high levels of soluble RANKL correlated with the degree of bone damage.	[77,78,79,80,81]
Growth factors	FGF-2	FGF-2 promotes cancer progression and angiogenic potential; FGF-2, along with IL-6, can increase the proliferation of myeloma cells.	Low FGF-2 levels are associated with a shorter MM progression time; bFGF G allele is associated with worse response to therapy.	[108,109,111,144]
	VEGF	VEGF promotes cancer progression and angiogenic potential.	VEGF is positively correlated with IL-20 levels and the link between those two parameters may be used as an indicator of the disease progression and angiogenesis process.	[110,145]
	HGF	HGF is involved in the regulation of cell proliferation and survival, but also has an anti-apoptotic effect on MM cells.	High level of HGF is associated with an unfavorable prognosis.	[120,121]
	PDGF-β	PDGF-β promotes cancer angiogenic potential.	PDGF-β may be an important indicator of the immune status of an advanced MM patient.	[123,124]
	Ang-2	Ang-2 promotes cancer angiogenic potential.	Ang-2 is a biomarker of angiogenesis; Ang-1/Ang-2 ratio may be useful in MM.	[125,126]
Interferon	IFN- 𝛾	IFN-𝛾 inhibits cell proliferation.	IFN-𝛾 has a positive effect on rituximab treatment.	[82,88]
Chemokines	CCL2 and CCL3	CCL2 and CCL3 affect the infiltration of macrophages into the bone marrow, as well as elevated polarization into TAM; CCL3 influences the development of bone disease in MM.	CCL2 and CCL3 are involved in the development of chemoresistance; CCL3 levels correlate with the activity and stage of the disease; high levels of CCL3 are associated with unfavorable diagnosis.	[40,136,141,142,143]

**Table 2 ijms-22-03587-t002:** Schematic interpretation of the correlation of different parameters in MM patients vs. healthy individuals.

Parameter	Level	Ref
Neutrophil-to-lymphocyte ratio (NLR)	MM > control group	[156]
Monocyte-to-lymphocyte ratio (MLR)	MM > control group	[156,164]
Platels-to-lymphocyte ratio (PLR)	MM < control group	[156,166]
Regulatory T lymphocytes (Tregs)	MM > control group	[179,180]
Circulating Plasma Cells	MM > control group	[181]
Junctional adhesion molecule-A (JAM-A)	MM > control group	[186,190]
Autophagic markers – Beclin-1, LC3	MM > control group	[196]
Programmed death ligand 1 (PD-L1)	MM > control group	[204,205,206]

## Data Availability

Not applicable.

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
