# Peer review of "Immunological Prognostic Factors in Multiple Myeloma"

_ijms, 2021, doi:10.3390/ijms22073587_

Round 1

Reviewer 1 Report

Multiple myeloma is an incurable cancer of plasma cells in the bone marrow. It is now well appreciated that inflammatory environment and immune regulation are essential factors contributing to disease occurrence, progression and drug resistance. Numerous studies are underway to better understand the immune system de-regulation in myeloma and develop therapies that can restore the altered immune environment and induce anti myeloma responses. The review by Bebnowska and colleagues is therefore in an important area of research in multiple myeloma and provides a comprehensive review on immunological factors in myeloma. However, I have several comments listed below that I feel will significantly improve the flow of information presented in the review.

There are several grammatical mistakes and incomplete sentences throughout the review that the authors must correct.

The section on cytokines could be sub-divided into two sections: pro-inflammatory and anti-inflammatory cytokines.

IL-1: The authors should briefly discuss clinical studies done using anti-IL1 therapies and the findings observed in that study.

IL-2: Again, clinical studies using IL-2 must be discussed.

IL-6: Clinical studies must be discussed in detail. If IL-6 is important for myeloma cells to grow and survive, the authors should explain why they think clinical trials using IL-6 antibodies showed modest results.

The authors should also review existing literature on a few other key chemokines and cytokines IL-8, IGF-1, IL-1RA, IL-4 and IL-12.

VEGF: The section on VEGF should be re-written. In the present version, the information provided is difficult to understand. Results from anti-VEGF clinical trials should be discussed.

Conclusion: I feel the title of this section should be changed to include “future directions”. The authors should briefly explain how the altered cytokine environment and cells of the immune system are currently targeted to improve disease prognosis in myeloma? In addition to existing approaches, what could one do different to further improve outcomes in myeloma?

Minor comment: Reference numbers 20 and 48 are the same.

Author Response

Dear Reviewer,

      On behalf of the Authors of our review paper entitled “Immunological prognostic factors in multiple myeloma” (IJMS-1112982) we would like to cordially thank you for the review. We tried to follow all your suggestions and corrected the manuscript accordingly. Here are the point-by-point answers, all changes in the manuscript are marked in red.

Multiple myeloma is an incurable cancer of plasma cells in the bone marrow. It is now well appreciated that inflammatory environment and immune regulation are essential factors contributing to disease occurrence, progression and drug resistance. Numerous studies are underway to better understand the immune system de-regulation in myeloma and develop therapies that can restore the altered immune environment and induce anti myeloma responses. The review by Bebnowska and colleagues is therefore in an important area of research in multiple myeloma and provides a comprehensive review on immunological factors in myeloma. However, I have several comments listed below that I feel will significantly improve the flow of information presented in the review.

There are several grammatical mistakes and incomplete sentences throughout the review that the authors must correct.

RE: The paper underwent English corrections by a native speaker.

The section on cytokines could be sub-divided into two sections: pro-inflammatory and anti-inflammatory cytokines.

RE: As requested, the section on cytokines has been divided into pro- and anti-inflammatory.

IL-1: The authors should briefly discuss clinical studies done using anti-IL1 therapies and the findings observed in that study.

RE: Clinical studies using anti-IL-1 therapies have been discussed in the IL-1Ra section.

IL-2: Again, clinical studies using IL-2 must be discussed.

RE: Clinical studies using IL-2 have been discussed in the IL-2 section.

IL-6: Clinical studies must be discussed in detail. If IL-6 is important for myeloma cells to grow and survive, the authors should explain why they think clinical trials using IL-6 antibodies showed modest results.

RE: Clinical trials of anti-IL-6 therapy have been discussed thoroughly, but we have mainly focused on the prognostic properties of each of the factors.

The authors should also review existing literature on a few other key chemokines and cytokines IL-8, IGF-1, IL-1RA, IL-4 and IL-12.

RE: The section on IGF-1, IL-1Ra, IL-8, IL-4 and IL-12 in MM has been added to the manuscript, indicating if the prognostic value of these elements of the immune system has been described.

VEGF: The section on VEGF should be re-written. In the present version, the information provided is difficult to understand. Results from anti-VEGF clinical trials should be discussed.

RE: The section has been re-written and the anti-VEGF trials have been discussed.

Conclusion: I feel the title of this section should be changed to include “future directions”. The authors should briefly explain how the altered cytokine environment and cells of the immune system are currently targeted to improve disease prognosis in myeloma? In addition to existing approaches, what could one do different to further improve outcomes in myeloma?

RE: The section has been renamed and the data suggested by the Reviewer are now added to the manuscript. Thank you for your valuable comment.

Minor comment: Reference numbers 20 and 48 are the same.

RE: Thank you for your very valuable attention. We improved the citation.

Again, we would like to thank you for your valuable comments to improve our manuscript. We are hoping that in its current form the manuscript will fulfill the requirements of IJMS.

On behalf of the Authors,

Paulina Niedźwiedzka-Rystwej

Reviewer 2 Report

This paper provides a good overview of the immunological prognostic factors in multiple myeloma. Please consider revising the manuscript to correct the language and style to make it more grammatically accurate.

I think that the manuscript of Bębnowska et al. reported the main features of the immune components which represent the prognostic value in MM patients.
In addition to english improvment i suggest to add a table showing the main functions, the role as prognostic value for each immune components in order to facilitate the reading and the understanding of the text. 

Author Response

Dear Reviewer,

      On behalf of the Authors of our review paper entitled “Immunological prognostic factors in multiple myeloma” (IJMS-1112982) we would like to cordially thank you the review. We tried to follow all your suggestions and corrected the manuscript accordingly. Here are the point-by-point answers, all changes in the manuscript are marked in red.

This paper provides a good overview of the immunological prognostic factors in multiple myeloma. Please consider revising the manuscript to correct the language and style to make it more grammatically accurate.

RE: We did our best to improve the English throughout the text, thank you.

I think that the manuscript of Bębnowska et al. reported the main features of the immune components which represent the prognostic value in MM patients.
In addition to english improvment i suggest to add a table showing the main functions, the role as prognostic value for each immune components in order to facilitate the reading and the understanding of the text. 

RE: Thank you for your valuable advice. The table was added to the manuscript.

Again, we would like to thank you for your valuable comments to improve our manuscript. We are hoping that in its current form the manuscript will fulfill the requirements of IJMS.

On behalf of the Authors,

Paulina Niedźwiedzka-Rystwej

Round 2

Reviewer 1 Report

The authors have modified the article to include information on more cytokines and chemokines. There are still several grammatical mistakes. I feel the authors have not carefully revised and reviewed the paper. As an example, please see the new text on IL-6. Sentences 148-153 and 153-158 are the same. The authors should proofread the manuscript carefully and correct grammatical and scientific mistakes.

Author Response

Dear Reviewer,

Thank you for your valuable comments. The paper underwent professional English language editing, carried out by MPDI editing services.

We hope the paper is acceptable in its corrected form.

Kind regards,

Paulina Niedźwiedzka-Rystwej